# Sensitivity to Sweet and Salty Tastes in Children and Adolescents with Type 1 Diabetes

**DOI:** 10.3390/nu15010172

**Published:** 2022-12-30

**Authors:** Beata I. Sińska, Alicja Kucharska, Katarzyna Czarnecka, Anna Harton, Agnieszka Szypowska, Iwona Traczyk

**Affiliations:** 1Department of Human Nutrition, Faculty of Health Sciences, Medical University of Warsaw, 27 Erazma Ciołka Street, 01-445 Warsaw, Poland; 2Department of Dietetics, Institute of Human Nutrition Sciences, Warsaw University of Life Sciences (WULS), 159C Nowoursynowska Street, 02-776 Warsaw, Poland; 3Department of Pediatrics, Medical University of Warsaw, 02-091 Warsaw, Poland

**Keywords:** sweet taste, salty taste, type 1 diabetes

## Abstract

Taste function impairment is observed in people with type 1 diabetes (T1D). It is most often related to sweet taste. It is associated with such factors as diabetic neuropathy, smoking, age, duration of the disease and a rigorous diet that eliminates easily digestible carbohydrates. The aim of the study was to compare sensitivity to sweet and salty tastes between healthy children and adolescents and children and adolescents with T1D. The study group consisted of children with T1D (n = 35), with at least 5 years of disease history, while the group of healthy children included 46 individuals selected in terms of age, gender and BMI. A study concerning the perception of sweet and salty taste was carried out with the use of the specific gustometry method (examining the recognition and assessment of the intensity of taste sensations, performing a hedonic assessment). Children and adolescents from both groups had trouble recognising tastes. Children and adolescents with T1D were more likely to recognise sweet taste correctly even at its lower concentrations compared to healthy individuals (*p* = 0.04). Salty taste was significantly more often correctly identified by healthy children compared to T1D patients (*p* = 0.01). Children and adolescents with T1D reported a stronger intensity of perceived tastes than healthy ones. No significant differences in perceived pleasure were noted at lower sucrose concentrations in any group. The intensity score was higher in individuals with T1D at higher sucrose concentrations. No significant differences occurred in the assessment of salty taste intensity. The hedonic scoring of solutions with higher concentrations of sweet taste was higher in people with T1D than in healthy ones, while salty taste was assessed neutrally. Children and adolescents with T1D were demonstrated to have some taste recognition disorders. Therefore, monitoring taste function in pediatric diabetic clinical practice seems relevant, as it may be associated with important implications for the intake of a particular type of food and for the development of eating habits and preferences.

## 1. Introduction

Type 1 diabetes (T1D) is an autoimmune systemic disease. It affects all systems of the body, leading to a large number of complications. Some of them are well understood, while others, such as impaired perception of taste (dysgeusia) and smell (dysosmia), still need to be explored. A decrease in taste functions was commonly observed in people with diabetes. It was most often related to sweet taste [1,2]. However, the impaired perception of taste was not considered a serious disability and patients were not systematically tested in this respect [3]. 

The fact of the occurrence of a decrease in the taste sensitivity threshold in people with T1D was found to be associated with the degradation of neurons by diabetic neuropathy [4]. Neuropathy may affect numerous neurons, including those responsible for transmitting impulses from taste buds to taste centers in the cerebral cortex (gustatory cortex). Damage to neurons occurs as a result of chronic hyperglycemia causing lesions in the nerves responsible for taste sensation (nerves VII, IX, X) and pathological changes in the capillaries supplying the peripheral nerves [5,6]. Smoking, age and duration of the disease may also be indicated as factors influencing the perception of taste.

Le Floch et al. demonstrated that 78% of 57 patients with diabetes had poorer sensitivity to taste compared to healthy subjects (n = 38). Loss of taste (ageusia) affected 6 people, and hypogeusia (impaired sensitivity to taste stimuli) was found in other patients. The sensation of taste deteriorated with age, smoking and higher glycemia [2]. 

So far, numerous studies have been carried out on the issue, showing that people with T1D had a clearly elevated threshold for sweet taste [7,8].

The observations made in those studies are crucial for understanding the difficulties in adhering to dietary recommendations by people with T1D. It seems that the concentration of glucose necessary to reach the threshold of sweet taste recognition determines dietary choices leading to an increased consumption of sweet food [9]. With regards to dietary recommendations for diabetes, great importance is attached to reducing the supply of easily digestible carbohydrates in the diet, including added and free sugars [10]. In this context, the implementation of such recommendations is extremely difficult in practice. Therefore, it is worth considering whether the reason for the difficulties in implementing dietary recommendations by people with T1D and practicing unhealthy eating habits is not related to the lowered threshold of taste sensitivity, including sweet taste. 

Due to the small number of studies on this issue among children and adolescents, the aim of this study was to compare the sensitivity to sweet and salty taste among children and adolescents with type 1 diabetes and healthy ones.

## 2. Materials and Methods

### 2.1. Subjects/Participants

The study included 81 participants aged from 8–18 years. The study group (diabetes group) consisted of 35 children and adolescents with T1D who were patients of the 2nd Department of Pediatrics, Diabetic Subdivision, the Independent Public Children’s Clinical Hospital in Warsaw. The control group (non-diabetes group) included 46 healthy children and adolescents (students at the school and kindergarten complex no. 3 in Nowy Dwór Mazowiecki). With regards to children and adolescents with T1D, the duration of the disease (at least 5 years) and the consent of the doctor were the inclusion criteria. Participation in the study was voluntary. In accordance with the requirements of the Declaration of Helsinki, all participants and their legal guardians had been informed about study aims, nature and methodology, and had agreed in writing to participate in the study before joining the study. The study was approved by the Bioethics Committee of the Medical University of Warsaw (approval number KB/59/2018).

### 2.2. Methods

In the case of healthy children and adolescents, the study was conducted on the premises of the school in the presence of a class teacher. With regards to children and adolescents with T1D, the study was carried out in the hospital in the presence of a doctor in order to ensure patient safety.

On the day of the test, distilled water solutions were prepared with the addition of flavoring substances, i.e., sucrose and sodium chloride. The substances were weighed on an analytical balance with an accuracy of 0.01 g, according to the principles of the Polish Committee for Standardization PN ISO 3972 [11]. A total of 6 samples were prepared for each taste, including 3 at concentrations below the taste recognition threshold and 3 above this threshold (Table 1). 

Immediately before the test, samples of the solutions were poured into 15-mL disposable cups. Each participant received 6 coded samples of sweet taste solutions and 6 coded samples of salty taste solutions. The participants were blinded to the type and concentration of substances dissolved in the samples. Each participant was to taste each sample, holding it in the mouth for approximately 5 s, and then spitting it out without swallowing. After tasting each sample, the mouth was washed with distilled water. There was a 15-min gap between tasting the sweet and salty samples. The samples were tasted in a fixed order (from the least to the most concentrated solution).

The study was conducted in a sensory assessment room arranged for this purpose in the morning, and participants were asked not to eat meals an hour prior to the study. It was not necessary for the subjects to enter the study on an empty stomach. The used sample solutions went down the sewage system, while empty plastic containers were disposed of by a municipal waste company. 

#### 2.2.1. Taste Recognition

Taste recognition was assessed using the specific gustometry method. 

The participants of the study determined the perceived taste (sweet, salty, sour, bitter, umami) or lack thereof. 

#### 2.2.2. Assessment of the Intensity of Taste Sensations

The second stage of the study was to assess the intensity of taste sensations. For this purpose, a scale of 10 cm in length was used, with the described endpoints being 0 for lack of taste and 10 for maximally intense taste. The distance from point 0 to the point marked by the volunteer, converted to points ranging from 0 to 10 constituted the test result. 

#### 2.2.3. Hedonic Assessment

The scale used was a 10-cm segment with the described endpoints: the left endpoint—maximally unpleasant taste (−5) and the right endpoint—maximally pleasant taste (+5). The middle point was described as neutral taste (0). The numerical result of the hedonic assessment was marked by the distance from the central point of the scale to the point marked by the respondent expressed in points ranging from −5 to +5. 

#### 2.2.4. Sociodemographic and Anthropometric Data

Additionally, data were collected on age, sex, weight and height. In healthy children and adolescents, the data were obtained from the school nurse, and in children and adolescents with T1D, from the patient’s records. The body mass index (BMI) was calculated based on the weight and height [12], which was subsequently interpreted based on the national BMI percentile charts (OLAF and OLA) [13] for female and male participants aged from 3–18 years. 

### 2.3. Statistical Analysis

Statistical analysis was carried out with Statistica 13.1 using descriptive statistics, cross tabulation and the Mann–Whitney U test. A significance level of *p* < 0.1 was assumed to be statistically significant.

## 3. Results

### 3.1. Characteristics of the Study and Control Group

Female participants constituted 56% of all study participants. The average age of the respondents was 13.7 ± 1.8 years. Children and adolescents from the study group were significantly younger (12.6 ± 2.3 years vs 14.5 ± 0.7 years, *p* = 0.001). The average value of the BMI index of the respondents was 20.0 ± 3.6 kg/m^2^, without significant differences between the groups (Table 2). In the group of children and adolescents with T1D, the average duration of the disease was 5.2 ± 2.1 years (min. 5 years, max. 10 years). 

### 3.2. Taste Recognition

The taste recognition part of the study showed that over half of the children and adolescents with T1D and healthy children and adolescents could not identify the taste of the solution samples. Regardless of the concentration of sweet taste solutions, 54.8% of children with T1D and 53.6% of healthy children and adolescents did not recognise sweet taste, while 57.1% and 40.2% of the examined children and adolescents did not recognise salty taste, respectively. Children and adolescents with T1D significantly more often recognised sweet taste compared to healthy children and adolescents (35.2% vs. 26.5%, *p* = 0.04). In the case of salty taste, it was significantly more often identified by healthy children and adolescents than children and adolescents with T1D (37.7% vs. 26.7%, *p* = 0.01) (Table 3).

Detailed analysis of the taste recognition of sucrose solutions revealed that children and adolescents with T1D more often correctly identified sweet taste at lower sucrose concentrations compared to healthy children and adolescents. At a sucrose concentration of 2.59 g/L, 25.7% of children and adolescents with T1D and only 4.4% of healthy children and adolescents correctly recognised the taste (*p* = 0.005), and at a concentration of 4.32 g/L, the respective percentages of recognition were 40.0% and 17.4% (*p* = 0.02) (Table 4). 

With regards the recognition of salty taste, a higher percentage of correct identification of the taste was observed in healthy children and adolescents compared to children and adolescents with T1D at all concentrations of sodium chloride solutions. Twice fewer children and adolescents with T1D (22.9% vs. 43.5%, *p* = 0.05) recognised salty taste at a sodium chloride concentration of 0.98 g/L (Table 5).

### 3.3. Assessment of the Intensity of Taste Sensations

When assessing the intensity of the sensation of sucrose solutions, both tested groups were characterised by a higher intensity of sensation with an increase in the concentration of samples. Children and adolescents with T1D declared a stronger intensity of perceived tastes than their healthy peers. However, no statistically significant differences were noted (Table 6).

When assessing the intensity of the sensation of sodium chloride solutions in both groups, an increase in taste sensations was observed along with an increase in the concentration of sodium chloride. No statistically significant differences occurred between the groups at all tested concentrations (Table 7). 

### 3.4. Hedonic Assessment

No significant differences were determined in the perceived pleasure between healthy individuals and those with T1D during the consumption of solutions with lower sucrose concentrations (from 0.94 to 2.59 g/L). The hedonic assessment score of solutions at the concentrations of 4.32, 7.20 and 12.0 g/L was higher in children and adolescents with T1D than in healthy ones. Statistically significant differences were found at the concentrations of 4.32 g/L and 7.20 g/L (*p* = 0.008 and *p* = 0.03, respectively) (Table 8). 

Neutral taste sensation was the highest score in the hedonic assessment of sodium chloride solutions, both among healthy children and adolescents and those with diabetes. The unpleasant taste sensation intensified in the participants with an increase in the concentrations of solutions. The assessment of the pleasure of experiencing salty taste was less negative among healthy participants compared to participants with T1D. However, statistical differences between the groups were not revealed (Table 9).

## 4. Discussion

Nutrition in type 1 diabetes is a very important element of behavioral therapy. It directly affects glycemic parameters, maintaining good metabolic control and high quality of life of patients. With regards to children and adolescents with T1D, the Polish Diabetes Association and the International Society for Pediatric and Adolescent Diabetes (ISPAD 2020) recommended a diet consistent with the guidelines for the general population [10,14], with particular attention paid to the quantity and quality of carbohydrates. Observations showed that those guidelines were often not implemented [15,16], which contributed to an increase in the risk of adverse health effects, such as excessive body weight, hypertension, cardiovascular diseases and other complications of diabetes [17]. 

The perception of taste is one of the most important factors influencing individual preferences and eating habits with possible consequences for health [18,19]. The sense of taste occurs when the concentration of a flavouring substance reaches a threshold level that activates taste receptors to produce functional potentials in nerve fibers that are strong enough to induce taste perception [20]. Taste impairment in diabetes may affect the ability to follow a diet and, thus, good glycaemic control. Most of the studies conducted so far and related to the discussed issue were carried out among adults. They showed that patients with diabetes were characterised by a lower ability to identify tastes compared to healthy people [2,9,21].

With regards to adults with diabetes, the taste may change physiologically with age and due to possible complications caused by the disease. Hyperglycaemia and its relationship with microvascular complications, such as polyneuropathy, is one of the possible factors which may underlie dysgeusia in diabetes [22]. Peripheral nerve damage may affect lingual nerves, leading to taste impairment [2]. However, studies conducted in people with diabetes without polyneuropathy also revealed taste impairment, which is why this mechanism was widely discussed [9,23,24]. Another possible cause of taste dysfunction observed in T1D may be due to an inflammation of the oral mucosa [25] and reduced saliva secretion [26]. The described taste impairment in adult patients with T1D was most commonly associated with reduced taste functions and the impairment of sweet taste perception [27,28]. The results obtained in the present study did not confirm the described observations. It may be due to the fact that children and adolescents participated in the study, i.e., people with a relatively short duration of the disease, who have not yet developed serious complications. Hartman et al. demonstrated a more frequent misrecognition and reduced intensity of the perception of the taste of sucrose solution in children with T1D compared to healthy children [6]; however, the participants were older and had a longer duration of the disease than the participants of the present study.

In addition, the present study showed that children and adolescents with T1D significantly more often correctly recognised sweet taste compared to healthy children and adolescents. They also more commonly correctly identified this taste at lower concentrations of sucrose solutions. When assessing the intensity of the sensation of sucrose solutions, both tested groups were characterised by a higher intensity of sensation with an increase in the concentration of samples. Children and adolescents with T1D declared a stronger intensity of perceived tastes than their healthy peers, however, without statistically significant differences. The hedonic assessment score of solutions at higher concentrations (4.32, 7.20 and 12.0 g/L) was significantly higher in children and adolescents with T1D than in healthy ones. Healthy participants enjoyed eating sucrose solutions only at the highest concentration, and the assessment was still lower than in participants with diabetes. A higher sensitivity to sweet taste found in children and adolescents with T1D may be explained by following dietary recommendations, which focus mainly on limiting the amount of easily digestible carbohydrates, i.e., sweet-tasting products. Children and adolescents who limit sugar in their diet on a daily basis may be more sensitive to its taste. Healthy children consume all products and usually do not control the number of sweets in the diet; they are used to a high sweetness threshold [1]. The analysis of diet presented by Juruć et al. revealed that children and adolescents with T1D consumed smaller amounts of carbohydrates than their healthy peers [29]. Greater restrictions on the supply of sugars in the diet, especially at the early stages of the disease, result from greater adherence to the diet by parents and children. People with T1D consume whole-grain products, vegetables, fish and eggs more often than healthy ones, and have better habits in terms of not snacking between meals or not eating sweets. Reduced supply of simple carbohydrates and regularity in eating meals may contribute to lowering the sensitivity of taste to sweetness, i.e., recognising sweetness at lower concentrations.

The mechanisms of the development of salty taste hypogeusia in people with diabetes are similar to those occurring in the case of sweet taste. However, only the role of changes in the expression of alpha-gustucin should be ignored because it is not involved in the transduction of salty taste. With regards to the recognition of salty taste, a higher percentage of correct identification of taste was observed in healthy children and adolescents compared to children and adolescents with T1D at all concentrations of sodium chloride solutions. When assessing the intensity of the sensation of sodium chloride solutions in both groups, an increase in taste sensation was observed along with an increase in the concentration of sodium chloride without intergroup differences. Neutral taste sensation was the highest score in the hedonic assessment of sodium chloride solutions, both among healthy children and adolescents and those with T1D. Unpleasant taste sensation intensified in the participants with an increase in the concentration of solutions. The assessment of the pleasure of experiencing salty taste was less negative in healthy participants compared to participants with T1D. However, statistical differences were not demonstrated between groups. Consistently with the results obtained in our own study, Hartman et al. confirmed the reduced sensitivity of salty taste, which resulted in a lower percentage of correct recognition of the taste of sodium chloride solutions in the group of children with T1D compared to the group of healthy children and a higher percentage of misrecognition of this taste in children with T1D compared to healthy children. Moreover, the study by Hartman et al. showed that children with T1D had a tendency for a less negative reception of the salty taste of sodium chloride than children from the control group [6], which was not confirmed in the present study. Khobragade et al. also demonstrated reduced sensitivity to salty taste in people with T1D. [21].

Taste impairment in children with T1D may be associated with the early onset of the disease. Catamo A. et al. revealed a relationship between impaired taste function and early T1D disease, regardless of the duration of diabetes [30]. Early onset of T1D may be associated with a higher number of autoimmune disorders, lower initial insulin reservoir and higher insulin demand a year after diagnosis contributing to faster progression of diabetes and the occurrence of complications, including polyneuropathic complications and taste impairment [31,32]. 

Data on the perception of basic tastes in people with T1D are limited and show inconsistent results. The differences may result from a large heterogeneity of the study group, including glycaemic control, disease duration, the presence of comorbidities, chronic complications and other possible confounding factors. Therefore, the continuation of research on this subject is very important, even when facing some limitations. They could not have been avoided in the present study either; hence, our results should be interpreted in the light of the limitations of the study. First of all, the results of our study cannot be generalised for all T1D children and adolescents because the studied population is not representative. It was recruited using convenient sampling. Moreover, caution should be taken when interpreting the results obtained because the potential relationship between the change in taste and T1D-related complications were not taken into consideration in the present study. In addition, we did not assess the impact of genetic polymorphism, oral microbiota and menstrual cycle in the female participants. However, an interesting issue that was not comprised in the paper would be to compare the sweet taste recognition thresholds in diabetics and different levels of HbA1c. Testing subjects and controls performed the tests in different environments. Both in the hospital and at school, there was a sensory assessment room arranged for this purpose, the method of sample preparation and testing was carried out according to the same study protocol. The participants were provided with a peaceful and quiet place for individual testing.

Despite the limitations, our study is one of the few conducted in our country, and, thus, may contribute to a better understanding of diet-related dependencies in the population at an increased health risk.

## 5. Conclusions

In conclusion, our study showed that children and adolescents with T1D were much more likely to recognise sweet taste than their healthy colleagues. A reduced reaction to sweet taste may have resulted from a stricter adherence to dietary recommendations at the initial phase of the disease, as described in the literature. Additionally, the group of children and adolescents with T1D were found to have a reduced sensitivity to salty taste. Therefore, monitoring gustation in pediatric diabetic clinical practice seems relevant, as it may be associated with important implications for the intake of particular types of food and, in the longer term, for the development of eating habits and preferences.

## Figures and Tables

**Table 1 nutrients-15-00172-t001:** Concentrations of flavour solutions used in the sensory test [g/L].

Concentrations of Solutions [g/L]	
Sucrose	Sodium Chloride
0.94	0.34
1.56	0.48
2.59	0.69
4.32 *	0.98 *
7.20	1.40
12.00	2.00

* taste recognition threshold.

**Table 2 nutrients-15-00172-t002:** Characteristics of the study group.

Variable		Diabetes Group *n* = 35	Non-Diabetes Group *n* = 46	Total *n* = 81	*p* *
Female participants [%]		57	54	56	0.80
Male participants [%]		43	46	44	
Age [years]	M ± SD	12.6 ± 2.3	14.5 ± 0.7	13.7 ± 1.8	0.001
	Me	12.8	14.7	14.1	
	Min-Max	8.6–17.2	13.4–16.3	8.6–17.2	
BMI [kg/m^2^]	M ± SD	19.3 ± 3.3	20.6 ± 3.8	20.0 ± 3.6	0.12
	Me	18.7	19.5	19.2	
	Min-Max	13.9–28.1	14.9–35.4	14.9–35.4	

M—mean, SD—standard deviation, Me—median, min—minimum value, Max—maximum value. * the Mann–Whitney U test differences between groups.

**Table 3 nutrients-15-00172-t003:** The overall evaluation of the recognition of solution tastes.

Taste Recognition	Diabetes Group	Non-Diabetes Group	*p*	Diabetes Group	Non-Diabetes Group	*p*
	Sweet Taste			Salty Taste		
Correct recognition (%)	35.2	26.5	0.04	26.7	37.7	0.01
Incorrect recognition	10.0	19.9	0.01	16.2	22.1	0.02
Lack of recognition	54.8	53.6	ns	57.1	40.2	0.0002

**Table 4 nutrients-15-00172-t004:** Evaluation of the recognition of the taste of sucrose solutions.

Sucrose Concentration [g/L]	Diabetes Group			Non-Diabetes Group			*p* *
	Percentage of Individuals [%]						
	Correct Recognition	Incorrect Recognition	Lack of Recognition	Correct Recognition	Incorrect Recognition	Lack of Recognition	
0.94	0.0	17.1	82.9	0.0	8.7	91.3	ns
1.56	11.4	14.3	74.3	4.4	23.9	71.7	ns
2.59	25.7	11.4	62.9	4.4	19.5	76.1	0.005
4.32	40.0	8.6	51.4	17.4	36.9	45.7	0.02
7.20	60.0	2.9	37.1	50.0	26.1	23.9	ns
12.0	74.3	5.7	20.0	82.6	4.4	13.0	ns

* Correct recognition diabetes group vs. non-diabetes group.

**Table 5 nutrients-15-00172-t005:** Evaluation of the recognition of the taste of sodium chloride solutions.

Sodium Chloride Concentration [g/L]	Diabetes Group			Non-Diabetes Group			*p* *
	Percentage of Individuals [%]						
	Correct Recognition	Incorrect Recognition	Lack of Recognition	Correct Recognition	Incorrect Recognition	Lack of Recognition	
0.34	14.3	22.8	62.9	15.2	41.3	43.5	ns
0.48	8.6	20.0	71.4	17.4	23.9	58.7	ns
0.69	14.3	20.0	65.7	21.7	30.5	47.8	ns
0.98	22.9	20.0	57.1	43.5	17.4	39.1	0.05
1.40	45.7	5.7	48.6	58.7	6.5	34.8	ns
2.00	54.3	8.6	37.1	69.6	13.0	17.4	ns

* Correct recognition diabetes group vs. non-diabetes group.

**Table 6 nutrients-15-00172-t006:** Evaluation of the intensity of the perceived taste of sucrose solutions by groups.

Group	Diabetes Group						Non-Diabetes Group					
Concentration of sucrose solutions (g/L)	0.94	1.56	2.59	4.32	7.20	12.00	0.94	1.56	2.59	4.32	7.20	12.00
M	0.9	1.9	2.8	3.9	5.3	6.7	0.6	1.6	1.8	3.1	4.1	6.2
SD	1.2	1.7	2.7	2.9	2.9	2.9	1.1	2.0	2.1	2.5	2.7	2.3
Q1	0	0.5	0	1	3	5	0	0	0.2	1	2	5
Q2	0.5	2	2	3	5	8	0	1	1	2.8	4	6.3
Q3	1	3	5	6	8	9	1	2	3	4	6.3	8

M—mean, SD—standard deviation, Q1—lower quartile, Q2—median, Q3—upper quartile.

**Table 7 nutrients-15-00172-t007:** Evaluation of the intensity of the perceived taste of sodium chloride solutions by group.

Group	Diabetes Group						Non-Diabetes Group					
Concentration of sodium chloride solutions (g/L)	0.34	0.48	0.69	0.98	1.40	2.00	0.34	0.48	0.69	0.98	1.40	2.00
M	2.6	2.8	3.6	5.2	5.2	6.7	3.6	2.9	3.6	4.2	5.5	6.7
SD	2.5	2.2	2.9	3.2	3.2	2.3	3.0	2.4	2.9	3.3	3.1	3.1
Q1	0.5	1	1	2	2	5.6	1	1	1	1.2	3.5	4.5
Q2	2	3	3	5	5	7.8	3	2.5	3	3.8	6	7.3
Q3	4	4	6	8	8	9	6	5	6	7	8	9

M—mean, SD—standard deviation, Q1—lower quartile, Q2—median, Q3—upper quartile.

**Table 8 nutrients-15-00172-t008:** Hedonic assessment of sucrose solutions by group.

Group	Diabetes Group						Non-Diabetes Group					
Concentration of sucrose solutions (g/L)	0.94	1.56	2.59	4.32	7.20	12.00	0.94	1.56	2.59	4.32	7.20	12.00
M	0.0	−0.2	0.0	0.7 *	1.5 **	1.9	−0.1	−0.4	−0.6	−0.7 *	0.1 **	0.9
SD	0.7	1.3	1.9	2.4	2.5	2.4	0.7	1.3	1.5	2.1	2.5	2.8
Q1	0	−0.5	−0.3	−0.5	0	0	0	−1	−1.5	−2.5	−1	−1
Q2	0	0	0	0	1.5	2	0	0	0	−0.5	0.1	1
Q3	0	1	1	2	4	4	0	0	0	1	2	3.5

M—mean, SD—standard deviation, Q1—lower quartile, Q2—median, Q3—upper quartile. * difference between diabetes and non-diabetes group (*p* = 0.008, the Mann–Whitney U test), ** difference between diabetes and non-diabetes group (*p* = 0.03, the Mann–Whitney U test).

**Table 9 nutrients-15-00172-t009:** Hedonic assessment of sodium chloride solutions by group.

Group	Diabetes Group						Non-Diabetes Group					
Concentration of sodium chloride solutions (g/L)	0.34	0.48	0.69	0.98	1.40	2.00	0.34	0.48	0.69	0.98	1.40	2.00
M	0.9	−1.4	−1.2	−1.8	−3.2	−3.7	−1.5	−1.3	−1.3	−2.2	−2.6	−2.4
SD	1.7	1.9	2.2	2.3	2.2	1.8	1.9	1.7	2.2	2.1	2.3	2.5
Q1	−2	−3	−3	−4	−5	−5	−3	−3	−3	−4	−4.5	−5
Q2	0	−0.5	−0.5	−1	−3	−4	−1	−0.8	−1.1	−3.0	−3.8	−4.0
Q3	0	0	0	0	0	−3	0	0	0	0	−0.5	−1

M—mean, SD—standard deviation, Q1—lower quartile, Q2—median, Q3—upper quartile.

## Data Availability

The datasets used and/or analysed during the current study areavailable from the corresponding author upon reasonable request.

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
