# Peer review of "Sensitivity to Sweet and Salty Tastes in Children and Adolescents with Type 1 Diabetes"

_nutrients, 2022, doi:10.3390/nu15010172_

Round 1
Reviewer 1 Report
The MS is on an important and interesting topic. The authors' work and dedication on working with children are commendable. I however have concerns about the age difference between the two groups - this factor must be included in the analyses to have convincing results.
Throughout the MS, children should be replaced with “children and adolescents” given the tested age range. Alternatively, an explicit definition should be listed. This might be important given that references about children and adolescents have been used in the Introduction.
Line 16 – T1D should be spelt in full in the first appearance.
Testing subjects and controls performed the tests in different environments – this factor should be considered when interpreting the results and explicitly discussed as a limitation.
Line 102 – Did the participants always do sweet first and then salt?
Did the tests happen in a group setting or individual setting?
The arrangements of section 2.2 is somewhat confusing. So the subjects tasted each sample, and then performed taste recognition, intensity and hedonic ratings? Did they only taste the sample once and then answered all of the questions? If this is the case, more details should be given about the order of the test, as well as the pace.
Section 2.3 is too brief. Can you please elaborate on what dataset each of these statistical model was applied?
Line 138: I prefer not to use the word “girls” – Female participants?
The significant age difference between the Test and Control group seems to be a major concern to this study. Was this factor being considered in the analyses?
Given the relatively large age range (as well as the significant age difference between the testing and control group), I recommend to the authors to perform sub-analyses based on break-down age groups. This may make a difference to understanding the results. For instance, over 50% of the children did not recognise sweet, could this be a reflection of language development across age?
Similarly, I feel not convinced about the results outlined in Line 152-153. Was age difference being considered in this analyses? Is it possible to add age as a covariate?
Analyses of intensity data in section 3.1 should definitely include age (gender and BMI as well) as covariates.
Line 214-220 – I feel this paragraph belongs to the Introduction than here.
Author Response
"Please see the attachment."

Reviewer 2 Report
There is a very clear and nicely wrote paper, very well structured and with results explained and well interpreted.
The authors intention to evaluate the impact of T1D on children taste is very good and useful for the comprehension of disease evolution and how this can be translated in life of children sick with T1D. A better understanding of such correlations will be very helpful to implement good eating habits and preserve children (later adults) health.
Congratulations for your study and efforts to help T1D children !
Author Response
Dear Reviewer,
Thank you very much for your review and appreciation of our work.
Best regards,
Authors